

# Seasonal variability of upwelling radiance polarization over the Southern Baltic surface

Włodzimierz Freda

Department of Physics, Gdynia Maritime University, Gdynia, 81-225, Poland

*Correspondence to:* Włodzimierz Freda (wfreda@am.gdynia.pl)

**Abstract.** Polarization of light may be used to improve the remote colour sensing of sea water. This can be done in a number of ways, such as limiting of sun glints, obtaining information about atmospheric aerosol properties for atmospheric correction as well as improving the interpretation of the water-leaving signal results. However polarization signals at the top of atmosphere (ToA), that include the water-leaving signal, is strongly influenced by atmospheric molecular scattering and by direct sun and

5 sky reflections from sea surface. For these reasons, it is necessary to better understand the factors that change the polarization of light in the atmosphere-ocean system. In this paper, the influence of seasonal variability of inherent optical properties (IOPs), wind speed and solar zenith angle (SZA) on the polarization of upwelling radiance over the sea surface is discussed. The presented results come from Monte Carlo simulations, which used averaged measurements of IOPs, collected for several years, as input data. The effects of simulations are presented in the form of polar plots of the upwelling radiance degree of polarization

(DoP). The results indicate that regardless of the wavelength and type of water, the highest value of the above water DoP is strongly correlated with the absorption-to-attenuation ratio. The correlation is a power function and it depends on both the SZA and the wind speed. The correlation versatility for different wavelengths is very unusual in optics of the sea and is therefore worth emphasizing.

*Copyright statement.* TEXT

## 1 Introduction

Polarization of light plays a large and still underestimated role in optical research of the ocean-atmosphere system. In marine water, it is often researched because some animals have vision that distinguishes the polarizing properties of incoming light (Cronin et al., 2003). It has long been known that some animals use polarization for navigation. The first discovery of this behaviour was documented by Frisch (1949), who examined the activity of honey bees. However, in an aquatic environment,

polarization vision focuses on signals and contrast enhancement mechanisms (Marshall and Cronin, 2011). The ability of certain marine animals, especially cephalopods, to see and analyse the polarized light has been known for many years (Moody and Parriss, 1960; Rowell and Wells, 1961). This ability was first discovered in octopuses. They can not only see polarised light, but also use the polarization analysis of a scene to choose places where they will not be visible. More surprising is that





cuttlefish and squids can produce some patterns of polarized light on their body surfaces to give signals to other animals of their species or to camouflage themselves against an aggressor (Shashar et al., 1996; Mäthger and Denton, 2001). These animals are able to control the reflection of polarized light by multilayer reflection cells in their skin (Crookes et al., 2004).

Although the biological aspects of the use of under-surface polarized light by animals seem to be very interesting, this article focuses on the polarization of above-surface upwelling radiance that can be used by oceanographers. They are interested in obtaining useful information by the remote sensing of ocean colour. This tool can obtain valuable information about large areas of the ocean in a short time. Obtaining so much information would not be possible by direct "*in situ*" measurements. However, the radiation measured at Top of Atmosphere (ToA) is dominated by light scattered in the atmosphere. For this reason, mea-
surements made at the ToA must undergo an atmospheric correction, which significantly depends on the aerosol properties. Zhai et al. (2017) showed that, in general, the polarized signal at the ToA is 2-3 times higher than its water-leaving part, because of the influence of molecular scattering in the atmosphere. Discussions on the use of remote polarization measurements to determine the aerosol properties that can be then used for atmospheric correction are included in Chowdhary et al. (2002); Mishchenko and Travis (1997) as well as Hasekamp and Landgraf (2005). While Pust et al. (2011) show that measurements of
the degree of polarization of the sky (made from the ground) can also be helpful in obtaining aerosol parameters.

       Another factor that disturbs the colour remote sensing is surface reflected light that comes both from sky reflections and sun glints, that for some directions makes it difficult to gain a signal from water depth. The way to improve the retrieval of the sea colour is to use polarization. For this purpose, He et al. (2014) proposed to measure the Parallel Polarization Radiance (PPR)
instead of radiance intensity. Liu et al. (2017) based on radiative transfer modelling and laboratory measurement, showed that the concentration of particulate matter influences the PPR measured at ToA. Reduction of sun glints to improve the ocean colour retrieval has been studied by Zhou et al. (2017) as well as Garaba and Zielinski (2013) and limitation of sky reflections by observation of sea surface at the Brewster angle from the ship board has been examined by Wood and Cunningham (2001) and by Cunningham et al. (2002). Polarization distribution of skylight reflected off the rough sea surface has been examined by
Zhou et al. (2013). They simulated Degree of Polarization (DoP) as well as the Angle of Polarization (AoP) for reflected parts of upwelling radiance and discussed its variability with SZA for $0^o$, $30^o$, $60^o$ and $90^o$. Moreover, Zhou et al. (2013) showed the influence of wind speed and direction on the polarization pattern of reflected radiance.

       The water-leaving part of radiance over the ocean is strongly influenced by water constituents. The influence of Inherent
Optical Properties on the polarization of water-leaving radiance for two types of water was tested by Zhai et al. (2017). They analysed waters dominated by phytoplankton of three different chlorophyll a concentrations and waters dominated by nonalgae particles.

       Scattering properties of seawater represented by single scattering albedo that affects the polarization of water-leaving radi-
ance was examined by Piskozub and Freda (2013). They used bubble layers of various concentration to simulate its impact





on the degree of polarization. They also concluded that polarization remote sensing should be performed on a plane tilted approximately $90^o$ from the solar azimuth angle to avoid sun glints. There are also other articles that show the impact of IOPs on polarization of light. Ibrahim et al. (2012) show that attenuation to the absorption ratio influences the polarization of upwelling radiance below the sea surface.

Radiative transfer simulations have shown the effect of marine suspensions on the polarization of light recorded above sea surface (see Chami et al. (2005); Chami (2007)). Further studies have also shown the possibility of knowing the composition of suspension, i.e. the ratio of mineral to organic suspension, using polarization properties of water-leaving radiance (see Gilerson et al. (2006); Chami (2007); Tonizzo et al. (2011).

The author's intent is to show the seasonal variability of polarization of the upwelling radiance above the sea surface. The polarization properties of such a radiance comes from both water depth and sea surface reflections, and they can be measured by polarization imaging camera (Freda et al., 2015).

## 2 Methods

Comparison of measurements of an upwelling radiance DoP over wind-roughened sea surface for various seasons is difficult in the Southern Baltic region. It is caused by small number of sunny days in winter and too many variable weather factors that would make it difficult to explain the differences. These undesirable weather factors are: different aerosol optical depths, covering the sky with clouds, different speeds and direction of wind relative to the position of the Sun. For those reasons, it was decided to compare the effect of seasonal changes on the polarization of upwelling radiance by Monte Carlo simulation.

Such a modelling is based on measurements of Inherent Optical Properties (IOPs) made in the Southern Baltic basin. But weather conditions like optical properties of aerosol and wave slopes caused by wind parameters are the same. For a detailed description of the inputs and conditions under which the simulation is performed, see the following subsections.

### 2.1 Measurements of IOPs in Southern Baltic

Inherent Optical Properties of Baltic seawater and its constituents have been in the spotlight for oceanographers for two
decades. In addition to regular measurements of depth profiles of absorption and attenuation coefficients, measurements have been made for various components of seawater. Kowalczuk (1999) and Kowalczuk and Kaczmarek (1996) studied seasonal and spatial variability of yellow substances. They found that the high absorption in spring and low in winter is due to the biological cycle as well as the seasonal variability of the inlet with river water and its mixing. Further studies of these substances, later called Colored or Chromophoric Dissolved Organic Matter (CDOM), were made in the Southern Baltic by Kowalczuk et al.
(2005a, b, 2006, 2010); Meler et al. (2016a). CDOM is the primary absorber in Baltic Sea (Kowalczuk et al., 2010), and its impact to total absorption coefficient for blue light can reach up to 80% (Kowalczuk et al., 2005a). For this reason, the search for a better correlation of spectral remote sensing reflectance $R_{sr}$ ratio with the absorption coefficient of CDOM is still ongo-





ing. Kowalczuk et al. (2005a) proves that the ratio of remote sensing reflectance for 490nm related to that of 590nm is the most accurate in estimating the absorption of CDOM for a wavelength of 400nm in the Baltic Sea.

Measurements of suspended matter IOPs, i.e. particle absorption and particle scattering coefficients, have been compared with biogeochemical characteristics of suspended matter such as concentrations of suspended particulate matter, particulate organic matter, particulate organic carbon and chlorophyll a (Woźniak et al., 2011). The impact of suspended particulate matter has been tested by Meler et al. (2016b) in different regions of Baltic Sea. They concluded, that absorption properties of non-algal particles undergo larger regional than seasonal variability. They achieve the lowest values for open Baltic and coastal waters while the highest are observed near river mouths. The correlation of mass concentration of the suspended particulate matter and the particulate organic matter with absorption or backscattering coefficients have been examined by (Woźniak, 2014). He also used simulated spectral ratios of remote sensing reflectance to find its correlation with the concentration of particulate matter.

Total IOPs measurements in relation to underwater imaging applications was discussed by Levin et al. (2013). They concluded that important imaging parameters such as contrast, signal-to-noise ratio, visibility range and spatial resolution may be computed when the attenuation coefficients are known.

In addition to the absorption and attenuation coefficients the volume scattering functions (VSF) were also measured in the waters of the Southern Baltic (Freda et al., 2007; Freda and Piskozub, 2007; Freda, 2012). These measurements are significant because there is still no commercially available instrument that can measure the VSF in a wide range of scattering angles. The prototype volume scattering meter, described by Lee and Lewis (2003), has been used here. The same instrument (characterised by an angular resolution of $0.3^o$ and the range of scattering angles from $0.6^o$ to $177.9^o$) has been also used by Chami et al. (2005) in the Black Sea and by Berthon et al. (2007) in the Adriatic Sea.

The inherent optical properties of Baltic waters may be also changed by oil pollution. The influence of oil-in-water emulsion on the absorption coefficient of seawater was researched by Otremba (2007) as well as Haule and Freda (2016), while the influence on scattering properties has been tested by Freda (2014). The consequence of changes in IOPs for remote detection of oil-in-water emulsion has been discussed by Otremba et al. (2013); Otremba (2016); Haule et al. (2017).

Despite the many different sources of data of the absorption and attenuation coefficients coming from the Southern Baltic, the current paper uses the data contained in Sagan (2008). These data come from the largest dataset of ac-9 measurements in the Southern Baltic that was published in a tabular form of average values, extreme values and standard deviations. The data was recorded during cyclical cruises aboard R/V Oceania in 1999 and 2003 to 2005. Measurements were made in different months of the year. The data set was divided into two seasons. The split was made because it was possible to distinguish the months in which strong phytoplankton growth was observed (the summer season) and the months of low biological activity





(the winter season). In addition, Sagan (2008) distinguished three types of seawater: open Baltic, gulfs (Gulf of Gdańsk and Pomeranian Gulf) and coastal waters.

5 Total absorption coefficient taken to the simulation is a sum $a = (a - a_w) + a_w$, where $(a - a_w)$ is an average of the N number of measured depth profiles (see tab. 1) made with the ac-9 after Sagan (2008), and $a_w$ comes from Pope and Fry (1997). Similarly, the total attenuation coefficient is defined as $c = (c - c_w) + a_w + b_w$. Where the component $(c - c_w)$ comes directly from Sagan (2008), but to get total attenuation coefficient it was enlarged by clean water components of absorption $a_w$ and scattering $b_w$ (Smith and Baker, 1981). According to Sagan (2008), the highest values of IOPs and their variabilities are observed for water of gulfs and mouths of rivers that are located nearby.

## 10 2.2 Monte Carlo simulation of DoP

A simulation has been made with Monte Carlo algorithm created by Jacek Piskozub and described in Piskozub and Freda (2013). This version of algorithm not only collects information about various events, in which virtual photons are involved, like reflection, refraction, scattering, absorption or leaving the atmosphere. But it also gathers information about polarization changes for each photon during these events. Such information is described by four elements of the Stokes vector:
$S = [I, Q, U, V]^T$. Where I is the total radiance of light, Q describes the radiance of linearly polarised light (vertical to horizontal), U describes the radiance of linearly polarised light (diagonal right-skewed to left-skewed), V describes the circular polarization (clockwise to counter clockwise) and T denotes the transpose. Three elements Q, U and V of Stokes vector may be both positive or negative. The single quantity that characterizes these properties is the degree of polarization (DoP):

$$DoP = \frac{\sqrt{Q^2 + U^2 + V^2}}{I} \quad (1)$$

The defined degree of polarization is often replaced by the degree of linear polarization DoLP.

$$DoLP = \frac{\sqrt{Q^2 + U^2}}{I} \quad (2)$$

The latter hardly differs from DoP, because circular polarization represented by the V element of the Stokes vector is negligible in sea water (see off-diagonal Mueller matrix elements in (Voss and Fry, 1984)).

A virtual light source sends randomly polarised photons and the probability of occurrence of each mentioned above event is determined by the appropriate coefficients, like scattering coefficients or absorption coefficients. Reflection and refraction events are described by Fresnel equations, but the slope of sea surface is characterised by the wind-dependent distribution of Cox and Munk (1956). Angular distribution of scattered light is described by phase functions that, for both atmosphere and sea depth, are characterized separately for molecular scattering and particle scattering. Polarization properties of particle scattering
is described by Müeller matrices that for sea water are taken from Voss and Fry (1984) and for atmospheric aerosol particles





**Table 1.** Total absorption coefficients, total attenuation coefficients and their ratios. Average values measured by Sagan (2008) in Southern Baltic in 1999 and 2003 to 2005

| Summer season | | | | | | | | | |
|---|---|---|---|---|---|---|---|---|---|
| $\lambda$ [nm] | Baltic, N = 930 | | | Gulfs, N = 1428 | | | Coastal waters, N = 132 | | |
| | $a(\lambda)[m^{-1}]$ | $c(\lambda)[m^{-1}]$ | $\frac{a(\lambda)}{c(\lambda)}$ | $a[m^{-1}]$ | $c[m^{-1}]$ | $\frac{a(\lambda)}{c(\lambda)}$ | $a[m^{-1}]$ | $c[m^{-1}]$ | $\frac{a(\lambda)}{c(\lambda)}$ |
| 412 | 0.595 | 1.230 | 0.483 | 1.095 | 2.560 | 0.428 | 0.615 | 1.470 | 0.418 |
| 440 | 0.396 | 0.999 | 0.396 | 0.786 | 2.209 | 0.356 | 0.416 | 1.249 | 0.333 |
| 488 | 0.214 | 0.817 | 0.262 | 0.444 | 1.887 | 0.236 | 0.214 | 1.027 | 0.209 |
| 510 | 0.183 | 0.785 | 0.233 | 0.353 | 1.825 | 0.193 | 0.173 | 0.995 | 0.173 |
| 532 | 0.155 | 0.756 | 0.205 | 0.285 | 1.756 | 0.162 | 0.155 | 0.946 | 0.163 |
| 555 | 0.140 | 0.731 | 0.192 | 0.240 | 1.711 | 0.140 | 0.140 | 0.911 | 0.153 |
| 650 | 0.380 | 0.921 | 0.413 | 0.430 | 1.811 | 0.237 | 0.380 | 1.071 | 0.355 |
| 676 | 0.518 | 1.029 | 0.503 | 0.638 | 1.909 | 0.334 | 0.508 | 1.169 | 0.435 |
| Winter season | | | | | | | | | |
| $\lambda$ [nm] | Baltic, N = 930 | | | Gulfs, N = 1428 | | | Coastal waters, N = 132 | | |
| | $a(\lambda)[m^{-1}]$ | $c(\lambda)[m^{-1}]$ | $\frac{a(\lambda)}{c(\lambda)}$ | $a[m^{-1}]$ | $c[m^{-1}]$ | $\frac{a(\lambda)}{c(\lambda)}$ | $a[m^{-1}]$ | $c[m^{-1}]$ | $\frac{a(\lambda)}{c(\lambda)}$ |
| 412 | 0.485 | 0.680 | 0.713 | 0.755 | 1.760 | 0.429 | 0.585 | 1.250 | 0.468 |
| 440 | 0.296 | 0.479 | 0.618 | 0.496 | 1.469 | 0.338 | 0.376 | 1.009 | 0.373 |
| 488 | 0.154 | 0.337 | 0.459 | 0.264 | 1.217 | 0.217 | 0.194 | 0.817 | 0.238 |
| 510 | 0.133 | 0.315 | 0.421 | 0.213 | 1.165 | 0.182 | 0.163 | 0.785 | 0.207 |
| 532 | 0.125 | 0.296 | 0.421 | 0.185 | 1.106 | 0.167 | 0.145 | 0.756 | 0.191 |
| 555 | 0.120 | 0.291 | 0.411 | 0.170 | 1.071 | 0.158 | 0.140 | 0.731 | 0.191 |
| 650 | 0.370 | 0.521 | 0.711 | 0.400 | 1.211 | 0.330 | 0.370 | 0.931 | 0.398 |
| 676 | 0.478 | 0.629 | 0.760 | 0.528 | 1.299 | 0.407 | 0.488 | 1.039 | 0.470 |

after **?**.

Solar zenith angles (SZA) in the Southern Baltic region strongly depend on the season. In months described by Sagan (2008) as summer, the highest sun position over the horizon, that means the minimum of SZA during sun culmination, varies between 31$^o$ in June (the longest day of year) and 58$^o$ in the end of September. For simulation, a single value of 45$^o$ was chosen as a summer SZA. For months of winter season, from October to the end of March, the minimum of SZA varies between 50$^o$ (end of March) and as high as 78$^o$ in December (the shortest day). Such values of SZA are reached around noon, and are higher in





the rest of the days. That is why SZA of $75^o$ was chosen as a representative of winter season.

The direction twisted by $45^o$ from the sun reflection plane and two speeds of wind of 5 m/s and 15 m/s were chosen arbitrarily. Wind speeds were used to obtain the distribution of sea surface slopes (Cox and Munk, 1956) during transmission
and reflection processes.

## 3   Results

Examples of simulation results are presented in Figures 1a and 1b. They form polar plots of DoP of upwelling radiance just above the sea surface. Figure 1a shows the degree of polarization for average IOPs values measured in the open waters of the Baltic Sea for a wavelength of 412nm in the summer season, while Figure 1b depicts an analogous case but for the winter
season. These two results are characterized by one of the highest values of the peak of DoP of 0.88 for summer and 0.84 for winter. The azimuth position of the sun is $0^o$ in all cases. In a further discussion, the value of the DoP peak, called max(DoP), and its angular location for different types of waters, for two seasons and for different spectral channels will be discussed. Another example of the obtained results is the polar plot of DoP of waters of Gulf of Gdańsk and Pomeranian Gulf, simulated for the spectral band of 555nm, for summer (Fig. 2a) and for winter (Fig 2b). These results are characterized by the lowest
values of max(DoP).

The maximum values of DoP presented in Figure 1a (summer season, open Baltic waters) are visible for azimuth angles close to $180^o$ (direction of reflected sun) and a zenith angle of about $55^o$ although the solar zenith angle is $45^o$, while the max(DoP) in Figure 2a (summer season, waters of gulfs) is visible for zenith angle of about $60^o$. In contrast to the summer season case, the maximum DoPs in the winter season are close to zenith angle of $48^o$ (Figure 1b) and $54^o$ (Figure 2b), while
SZA is $75^o$. The lower position of the sun resulted in a higher position of the maximum DoP of upwelling radiation than its reflection angle, although a higher position of the sun resulted in a lower position of the maximum DoP. Another interesting effect is the higher DoP observed for directions close to the incident rays of sun (azimuth $0^o$). In the winter season, the values of DoP are higher than in summer, and zenith angles of this effect are lower in the winter than in the summer season.

In addition, to explain the differences between the summer and winter seasons, spectral radiances (in units of $Wm^{-2}sr^{-1}nm^{-1}$) have been calculated. Because of their high angular variability, they are presented as decimal logarithms of the spectral radiances (see Figures 1c and 1d). In Figure 1c there is log(I) for open waters in the 412nm channel in the summer season, (whose DoP is presented in Fig. 1a) and in Figure 1d there is log(I) for winter season (whose DoP is presented in the Fig. 1b). One can see that small SZA of summer season ($45^o$) resulted in low values of upwelling radiance that is stretched from direct reflection
point to the horizon, where is extended both left and right from azimuth of $180^o$. The high SZA of the winter season ($75^o$) resulted in much higher values of reflected light, which are also stretched from reflection point to the horizon.





**Table 2.** Parameters of equation (3), which describes the power trend lines in Figures 4a and 4b

| Simulation conditions | A | B | $R^2$ |
|---|---|---|---|
| SZA $45^o$, wind speed 5m/s | 1.102 | 0.262 | 0.973 |
| SZA $45^o$, wind speed 15m/s | 0.997 | 0.250 | 0.996 |
| SZA $75^o$, wind speed 5m/s | 0.903 | 0.117 | 0.906 |
| SZA $75^o$, wind speed 15m/s | 0.914 | 0.173 | 0.990 |

Moreover, as an attempt to explain the differences between seasons, the DoPs of underwater upwelling radiances were simulated. The results of such a simulation are shown in Figures 3a and 3b and the corresponding values of the radiance itself are presented in Figures 3c and 3d. These plots indicate that directions characterized by high values of DoP creates a dispersed ring, which is tilted from the horizontal direction. The angular tilt of the ring's plane is approximately $60^o$ for summer season and $45^o$ for the winter. Moreover, both the underwater upwelling spectral radiance and its DoP are higher in the summer season than in the winter.

All the values of maximum DoP of above-water upwelling radiance, for each absorption-to-attenuation ratio are collected in Figure 4. The summer season case is depicted in Figure 4a while the winter case in Figure 4b. The water types are marked with different symbols and two wind speeds are marked with different colours. The correlations of the highest DoP to the ratio of $a(\lambda)/c(\lambda)$ are almost proportional, however, correlation coefficient analysis has shown that the power functions are better matched. The trend line for the plot depicted in Figure 4a shows the relation of the maximum of DoP to the ratio of $a(\lambda)/c(\lambda)$ for the summer season and Figure 4b for the winter season. The presented trend lines may be described by power functions:

$$max(DoP) = A * \left( \frac{a(\lambda)}{c(\lambda)} \right)^B \tag{3}$$

whose parameters are collected in Table 2. It is worth emphasizing that these correlation are obtained for various spectral channels, hence they are wavelength independent for the examined spectral range.

An analysis of all collected data shows that higher values of maximum of DoP values are observed for lower wind speed. Moreover, max(DoP) have a higher range of variations in the summer season than in the winter. In Fig. 4a (summer season) one can see that values of max(DoP) are between 0.64 and 0.91 for a wind speed of 5m/s and between 0.61 and 0.83 for wind of 15m/s. However, in Fig. 4b (winter season) they are between 0.73 and 0.9 for wind speed of 5m/s and between 0.66 and 0.87 for wind of 15m/s. However, accurate data analysis shows that for the same wind speed the power trend lines for summer and winter seasons (in the Figures 4a and 4b) intersect. For a wind speed of 5m/s, the value of $a(\lambda)/c(\lambda)$ equal to 0.26 both power functions reach the same value of 0.77. For a wind speed of 15m/s, a value of $a(\lambda)/c(\lambda)$ equal to 0.32 both power functions reach the same value of 0.75. For lower absorption-to-attenuation ratios, winter DoPs have higher values than summer and for higher $a(\lambda)/c(\lambda)$ values the summer DoPs are higher.




## 4   Discussion

Polarization patterns of upwelling radiance, combined with reflected off sea surface and water-leaving, differs from those describing only the reflected part (Zhou et al., 2013). They show, that the DoP of reflected radiance component creates the ring at a polar plot called the Brewster zone. The angular shape of such a zone strongly depends on SZA (see Fig. 10 in Zhou et al. (2013)). However, the results of simulations presented here are similar to measurements of above water DoLP of upwelling radiance presented by Freda et al. (2015). Their DoLP results (see Figures 1 and 2 in (Freda et al., 2015)) are smaller than the results of simulated DoP. However, the difference may be caused by unknown environmental parameters during measurements, like high absorption coefficient in the waters of the river mouth, different aerosol optical depth other parameters. Despite the differences in the maximum of degree of polarization, the angular distribution has very similar pattern, with the peak in the vicinity of sun reflection azimuth angle.

The analysis of individual spectral bands shows that the highest values of absorption-to-attenuation ratio that correspond to the highest DoP occur in the case of both the smallest and the highest wavelengths. High values of absorption coefficient for small wavelengths is caused mainly by CDOM (Kowalczuk et al., 2005a). While a high absorption coefficient for high wavelengths (in the red spectral region) is caused by pure water (see (Pope and Fry, 1997)). The lowest values of max(DoP) for each type of water and for each season are observed for 555nm spectral band. The smallest values of absorption and weak spectral variability of scattering coefficient implies that the wavelength of 555nm is characterised by the lowest absorption-to-attenuation ratios. This is due to the existence of a minimum of absorption for water containing phytoplankton. Algae cells, depending on the composition of their pigments, may have a minimum of absorption in a wide range of spectral bands from 550nm to 660nm (Bricaud et al., 2004). But taking into account absorption of water that grows with wavelength (Pope and Fry, 1997), the minimum is in the spectral band of 555nm.

An analysis of water types show that the highest values of max(DoP) for each season were observed for open Baltic Sea water. The total $a(\lambda)/c(\lambda)$ ratio is higher in the open Baltic water because of the low scattering coefficients (Sagan, 2008). The value of the latter is determined mainly by the concentration of suspensions, which in open waters is significantly lower than in gulfs or coastal waters. According to Sagan (2008), the average particle scattering coefficient does not strongly depend on wavelengths and in open Baltic waters in winter season it varies between 0.15 (for 676nm) and 0.19 (for 412nm). While in the same season, but in waters of gulfs, the average particle scattering coefficient varies between 0.77 (for 676nm) and 1.00 (for 412nm).

Looking for the influence of water type on DoP, one can find that regardless of the wavelength and wind speed, the lowest max(DoP) values in winter are observed in waters of gulfs. These waters are characterised by the highest scattering coefficients, because of the high inflow of particle matter with river waters. But in the summer season, the lowest peak of DoP is observed for spectral bands from 412nm to 532nm in coastal waters, and in wavelengths from 555nm to 676nm in gulfs. The values of





a(λ)/c(λ) for these types of water are similar (for most wavelengths, the differences are smaller than 5%).

According to Fresnel equations, the reflections of two polarised components (parallel and perpendicular to the transmission/reflection plane) from the sea surface depend on two parameters only. They are the relative refractive index of medium

and the angle of incidence of light beam. Thus, it does not depend on the IOPs of seawater. The observed differences in polarization of upwelling radiance above sea surface for various absorption and attenuation coefficients of seawater come from the water-leaving component of that radiance, not from the reflected part. The reason for the correlation of the maximum DoP with the absorption-to-attenuation ratio is an occurrence of multiple scattering in water depth. The multiple photon scattering loses polarization with each occurrence. A high absorption-to-attenuation ratio means simply low scattering-to-attenuation impact

and, hence, little penetration of light in the water column and low participation in multiple scattering. Such a conclusion is in accordance with Piskozub and Freda (2013), who examined the influence of single scattering albedo on the polarization of water-leaving radiance. Their results show that in the sun reflection plane, the highest value of DoP is observed when total scattering coefficient is the lowest (see Fig. 3 in Piskozub and Freda (2013)).

The influence of wind speed on the DoP values shown in the Figure 4a and 4b is very clear. High wind speed simply means lower values of max(DoP). Such regularity is filled for all types of water and all spectral bands.

Although it is not clearly visible, the results of correlation of maximum DoP to the absorption-to-attenuation ratio seem to be coincident with the results of Ibrahim et al., (2012). Their correlation of c/a to DoLP seems to have a hyperbolic-like shape

(see Fig. 5 in Ibrahim et al. (2012), so for an inverted ratio of a/c their relation should be more-or-less linear.

The angular position of high polarization radiance for both below and above sea surface is schematically shown in the Figure 4. For a plane including sun reflection and refraction (azimuth $0^o$ and $180^o$) if the surface is calm, then for a SZA of $45^o$, the transmission angle TA, according to Snell's law, is $32^o$. Underwater polarization, caused by molecular scattering, is the highest

in the plane perpendicular to direction of propagation. In the vertical plane including transmitted rays, the highest polarization of molecular scattering (HPM) is $58^o$. For SZA of $75^o$ the transmission angle is $46^o$ and direction of HPM is $44^o$. Directions of HPM predicted for calm sea surface proved to be in the same directions for a wind-roughened sea surface (see the polarization patterns in the Figure 2a and 2b). The angular position of HPM, which is perpendicular to TA in the reflection-refraction plane, is both predicted by theory (molecular scattering) and confirmed by measurements of Tonizzo et al. (2009). The results pre-

sented by them come from waters characterised by various optical properties and a depth of one meter. Moreover, the authors mention the occurring clouds. These factors could be the reason why their underwater DoPs do not exceed the value of 0.4, and the measured angle is close to $100^o$ (see their Fig. 9).

The highest polarization of reflected radiance (HPR), according to Fresnel equations for flat reflective surface, should appear

if reflection angle is perpendicular to TA. The flat reflecting surface means that the angle of reflection is equal to the angle of



incidence, the latter is called the Brewster angle in this case. Figures 1a and 1b show that for a waved sea surface, the HPR direction is not equal to the angle of incidence (SZA). For SZA of $45^o$ (summer season case) the HPR is dispersed in an area close to $62^o$ (see Fig. 1a). The direction perpendicular to TA for a calm surface is $58^o$. Moreover, the sum of SZA and zenith angle of HPR is $107^o$, which is very close to double the Brewster angle, which for water is about $53^o$. That could explain the

5   angular position of HPR. However, for SZA of $75^o$ the HPR is dispersed in directions close to a zenith angle of 48? (see Fig. 1a). Here, the direction perpendicular to TA is $44^o$ (see Fig. 1b). The sum of SZA and HPR is $123^o$, which is much higher than double the Brewster angle. For this reason, the position of the HPR for rough sea surface should be interpreted rather as the direction perpendicular to the average TA, rather than the reflection from these surface slopes for which the angles of incidence and reflection are both equal to the Brewster angle.

The unpolarised sunlight after reflection or refraction processes at the sea surface will become partly polarised. Using Fresnel equations allows one to calculate both the transmission and reflection of two polarization components of that radiance. For a calm surface and a SZA of $45^o$ and TA of $32^o$, only 0.3% of the parallel polarised light component is reflected by the surface and 99.7% is transmitted to the water depth, while for the perpendicular polarization part much more (5.5%) is reflected and

94.5% is transmitted through the surface. For a SZA of $75^o$ (winter season) and a TA of 46? according to Fresnel equations, the reflections would be 11% for the parallel polarization component and as much as 32% for the perpendicular component, while the transmission would be equal to 89% and 68% respectively. These values show that for SZA of $45^o$ the radiance of reflected part of upwelling light is much lower than for SZA of $75^o$, but the contribution of one component of polarization (perpendicular) is much higher than the other, this would result in a higher degree of polarization of the reflected part. Moreover, these

theoretical values in a limited way reflect the high differences between values of upwelling radiance over sea surface shown in the Figures 1c and 1d.

Low transmission of light into water depth in winter case (SZA of $75^o$) in comparison to summer case (SZA of $45^o$) also explains the differences in underwater spectral radiances (see Figures 3c and 3d). Underwater radiance and its polarisation has

a higher influence on polarization of upwelling radiance over sea surface in the summer than in the winter. Lower SZA causes higher transmission into water depths and higher scattering towards the sea surface. This is why DoP distribution of upwelling radiance in winter is more like a polarization pattern of the reflected part presented by Zhou et al. (2013) with their ring-shaped Brewster zone.

## 5   Conclusions

The average values of total absorption and attenuation coefficients, measured in Southern Baltic waters were used to simulate the polarization properties of upwelling radiance over a waved sea surface. The degrees of polarization were simulated for all directions of the upper hemisphere. The pattern of DoP is similar to that measured by Freda et al. (2015). The results revealed the differences of DoP for separate seasons, types of water, speed of wind and wavelength. The presented analysis





focuses on the values and angular position of the highest DoP values. The main conclusion is that the highest values of DoP are almost proportional to the absorption-to-attenuation ratio. A thorough analysis has shown that the correlation depends on both solar zenith angle and wind speed, but is wavelength independent. Such a correlation is described by a power function whose parameters are given in Tab. 2. The directions of max(DoP) of combined water-leaving and reflected components of upwelling
5    radiance are dispersed in directions perpendicular to the under-surface transmission angle.

*Acknowledgements.* The author is grateful to prof. Sławomir Sagan and prof. Jacek Piskozub for their valuable comments and suggestions. The research presented in this paper was supported by grant No. UMO-2012/07/D/ST10/02865, funded by the National Science Centre (NCN) of Poland.



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



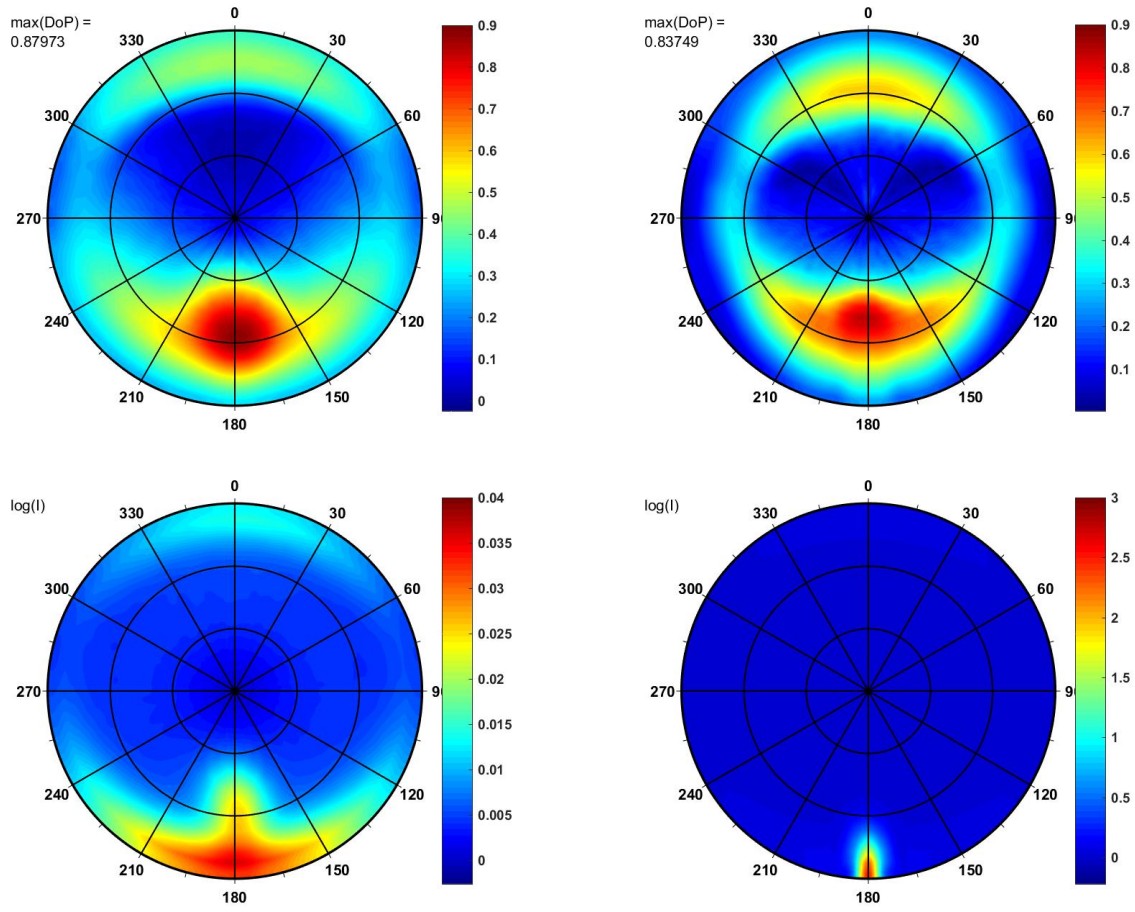

**Figure 1.** Simulation results of above-water upwelling radiance for average IOPs of open waters of the Southern Baltic, wavelength 412nm: a) DoP in summer season, SZA $45^o$, b) DoP in winter season, SZA $75^o$, c) decimal logarithm of upwelling radiance in summer season, SZA $45^o$, d) decimal logarithm of upwelling radiance in the winter season, SZA $75^o$





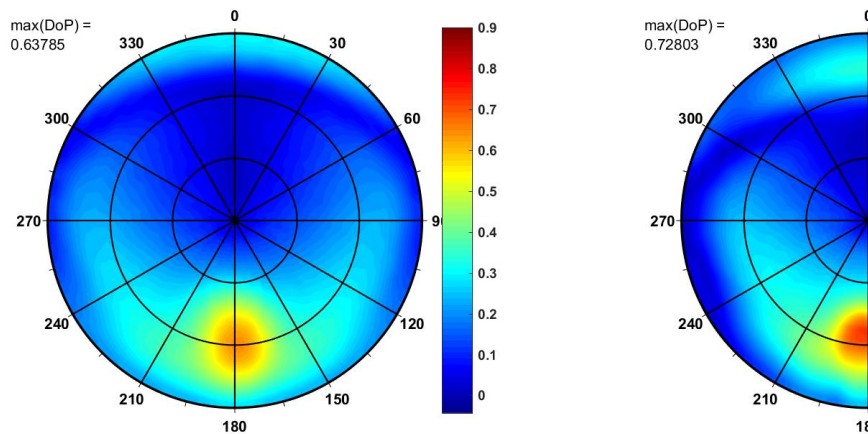
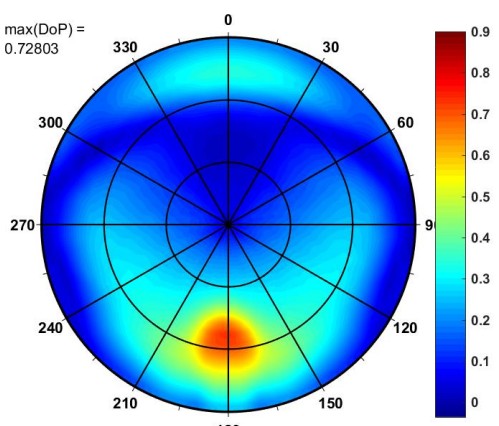

**Figure 2.** Degree of polarization of above-water upwelling radiance for average IOPs of gulf waters, wavelength 555nm: a) summer season, SZA $45^{o}$, b) winter season, SZA $75^{o}$. (the lowest DoP)





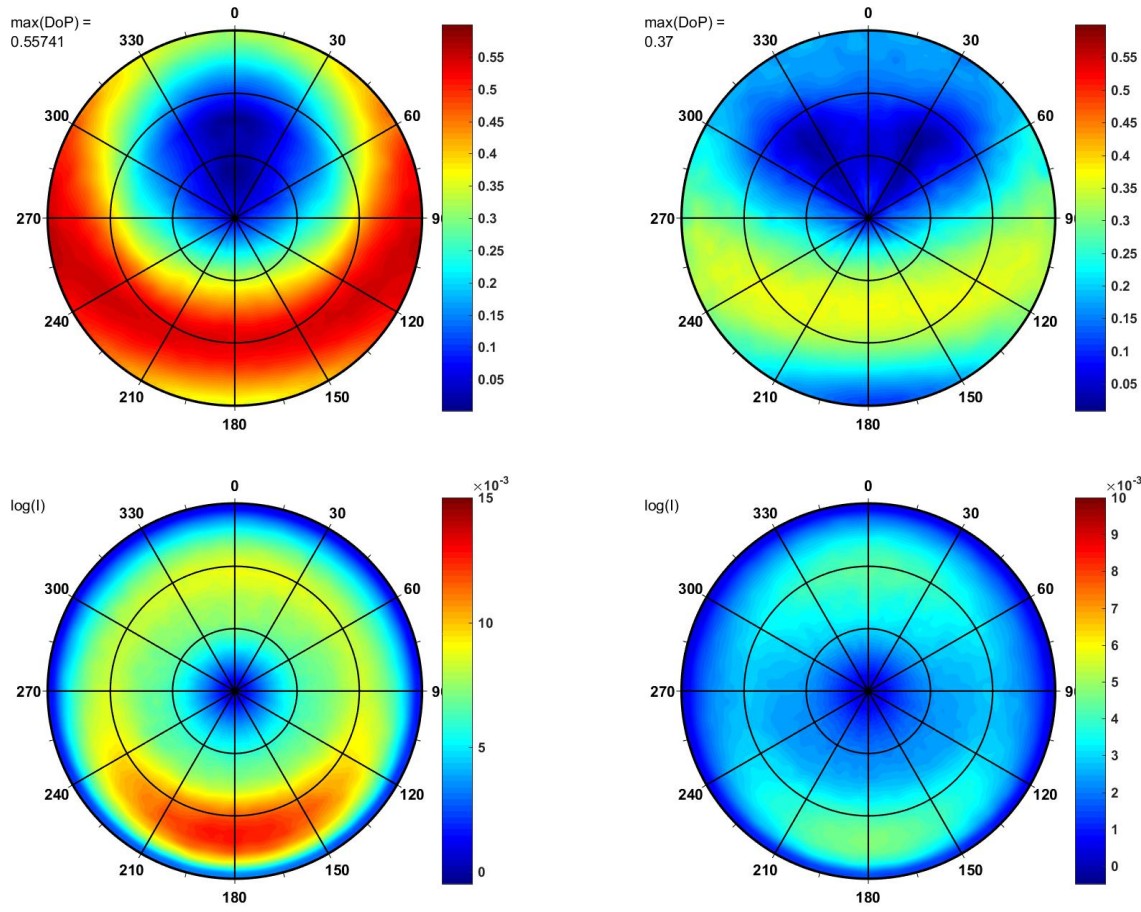

**Figure 3.** Simulation results of under-water upwelling radiance for average IOPs of open waters of the Southern Baltic, wavelength 412nm: a) DoP in summer season, SZA $45^o$, b) DoP in winter season, SZA $75^o$, c) decimal logarithm of upwelling radiance in summer season, SZA $45^o$, d) decimal logarithm of upwelling radiance in winter season, SZA $75^o$.





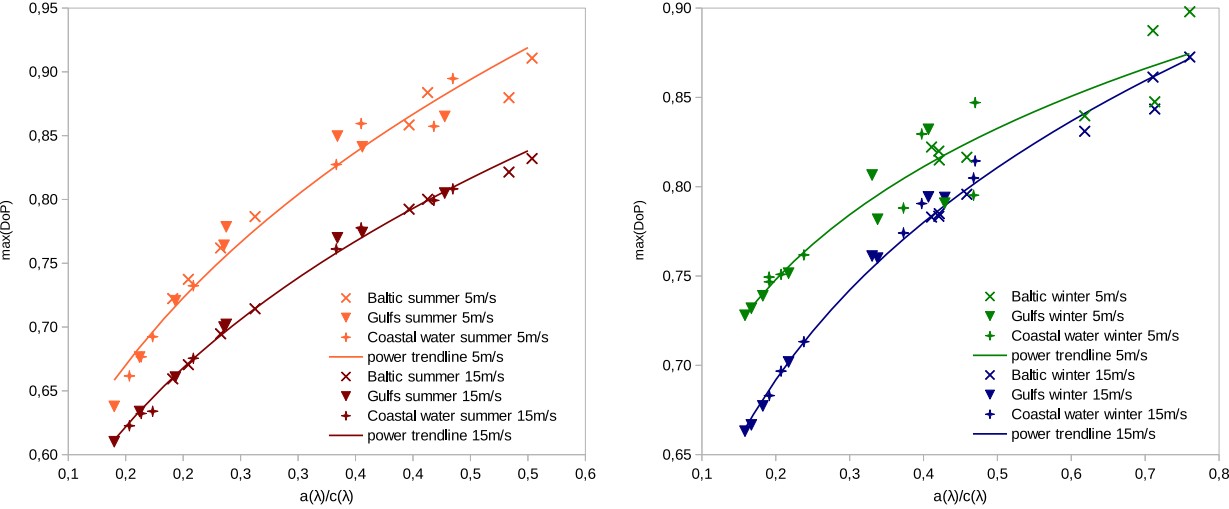

**Figure 4.** Maximum of degree of polarization against absorption-to-attenuation ratios a(λ)/c(λ) for average values of IOPs presented in Tab. 1, plotted for a) summer season, and b) winter season.

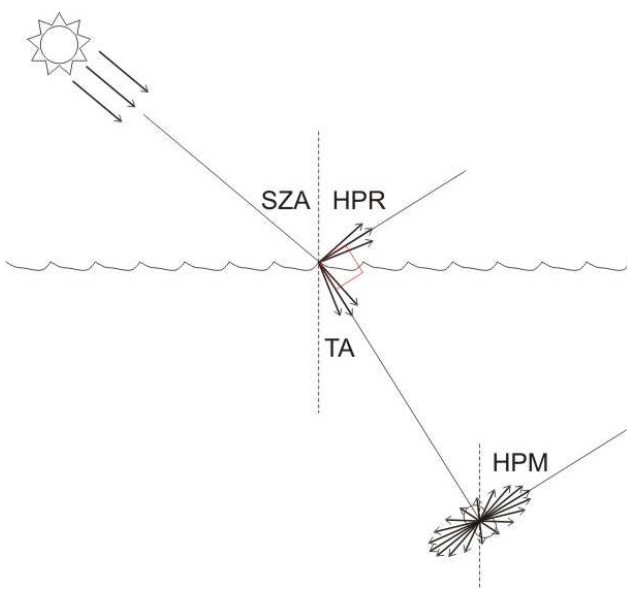

**Figure 5.** Directions of high polarization radiance for surface reflections (HPR) and for molecular scattering (HPM)