# Peer review of "Seasonal variability of upwelling radiance polarization over the Southern Baltic surface"

_Ocean Science, 2017_

## Referee Comment (RC1) · Anonymous Referee #1 · 8 Mar 2018

Although the addressed topic of polarization of upwelling radiance just below and just above the sea surface has scientific merit and significance, in my opinion this manuscript requires extensive major improvements in language and presentation of scientific content to make it suitable for a review process. In addition to numerous problems related to language, there are also various problems associated with unmet standards of scientific writing, such as the lack of precision in writing, the use of mental shortcuts, unclear formulation of scientific messages, inadequate preparation of figures (e.g., the lack of labels a,b, ... for the specific figure panels), and unsatisfactory figure captions that should be self-explanatory. For example, in Abstract that should be self-explanatory, the term "inherent optical properties" is used without indicating that it relates to seawater (and not to the atmosphere which could potentially be also

relevant in the context of Abstract), the term "wavelength" is used without indicating that it relates to light (and not, for example, to sea surface waves which could also be potentially relevant in the context of Abstract because sea surface is mentioned a few times), and the term "Monte Carlo simulations" is used without indicating what is actually being simulated (it should be specifically indicated that these simulations are for radiative transfer in the atmosphere-ocean system). It appears that a preparation of improved manuscript would benefit from involvement of an English language expert in scientific writing who can help achieve correct and improved language and meet the standards of scientific writing and presentation.

---

## Referee Comment (RC2) · Anonymous Referee #2 · 21 Mar 2018

The manuscript presents a very important topic in ocean color remote sensing and uses an extensive dataset combined with Monte carlo simulations. In my opinion the manuscript requires some revision to improve flow of text and aims of the study. Here are some aspects that could help improve the manuscript

General questions after reading the manuscript i am wondering -would polarization be useful in winter times assuming more days of overcast skies and diffuse light?

-what are the uncertainties in your observed dataset?

-i am missing the statistical analysis of the data in Table 1 to show the 'seasonal variability' of the dataset or at least the simulated dataset in my opinion since the goal of remote sensing is to get Remote sensing reflectance an essential climate variable, i

am wondering would it not be possible for your study to derive Rrs from the insitu (assuming this will be your reference dataset) measurements assuming a Bb etc then you also do the same for simulated polarized light to estimate the Rrs and compare the two Rrs from simulation and insitu looking at differences over seasons

-the IOPs of the different water types would vary seasonally and i am wondering do you have any additional data to support some of the points you raise in the discussion that is measurements of the water constituents and backscattering maybe even from satellite as a proxy. e.g Primary production enhanced or algal bloom could lead to higher scattering of the target region in the open water

-what is the message from Table 2 how can one use such information? what are the implications of this study fidnings in advancing ocean color remote sensing

additional comments Abstract line 1 -revise 'Polarization of light may be used to improve the remote colour sensing of sea water.', is it not something like 'Polarization of light leaving the ocean surface has the potential to improve quality of information retrieval from ocean color remote sensing (OCRS)'

line 2 -'sun glints' do you mean 'surface reflected and sky glint'

line 8 -'several years' can you put a number to this?

line 11 -'strong correlated' can you put some numbers to this as this is a qualitative statement

Introduction line 16 -replace 'marine' with 'aquatic' to include all water types

paragraph 1 -there is an interesting paper by Brady might be of interest to you Brady et al 2015. Open ocean fish reveal an omnidirectional solution to camouflage in polarized environments. Science. 350(6263):965-969

paragraph 3 -replace 'colour remote sensing' with OCRS ->ocean color remote sensing

line 17 - what do you mean by this sentence 'Another factor that disturbs the colour

remote sensing is surface reflected light that comes both from sky reflections and sun glints, that for some directions makes it difficult to gain a signal from water depth.' -is it not the case that surface reflected and sky glint affect OCRS –> because they are part of total water leaving radiance (glint + water constituents)?

line 2 page 6 - 'after ?.' what does this mean?

———————————————

---

## Author Comment (AC1) · 16 Apr 2018

The comments of the reviewer are quoted below, the replies are given under each comment.

Q1: Although the addressed topic of polarization of upwelling radiance just below and just above the sea surface has scientific merit and significance, in my opinion this manuscript requires extensive major improvements in language and presentation of scientific content to make it suitable for a review process. In addition to numerous problems related to language, there are also various problems associated with unmet standards of scientific writing, such as the lack of precision in writing, the use of mental shortcuts, unclear formulation of scientific messages, inadequate preparation of fig-

[Figure]

ures (e.g., the lack of labels a,b, ... for the specific figure panels), and unsatisfactory figure captions that should be self-explanatory.

Reply I am grateful that the reviewer appreciates the importance of this topic. I hope that the following amendments will make the article meet the correct form and standard for publication. I decided to publish in Ocean Science because Copernicus Publications applies typesetting and language copy-editing during production. Figure captions and labels are corrected.

Q2 For example, in Abstract that should be self-explanatory, the term "inherent optical properties" is used without indicating that it relates to seawater (and not to the atmosphere which could potentially be also relevant in the context of Abstract), the term "wavelength" is used without indicating that it relates to light (and not, for example, to sea surface waves which could also be potentially relevant in the context of Abstract because sea surface is mentioned a few times), and the term "Monte Carlo simulations" is used without indicating what is actually being simulated (it should be specifically indicated that these simulations are for radiative transfer in the atmosphere-ocean system).

Reply Thank you very much for these comments. I think that all of them are right and I have made a correct amendment to the final version of the article. See the abstract: lines 7, 8, 10 and 13.

Q3 It appears that a preparation of improved manuscript would benefit from involvement of an English language expert in scientific writing who can help achieve correct and improved language and meet the standards of scientific writing and presentation.

Reply I tried to prepare the article as best as possible, but I agree that an English language expert in scientific writing will help in obtaining a higher standard of the paper. That is why I give my permission for the language copy-editing before the final publication.

[Figure]

Please also note the supplement to this comment:
https://www.ocean-sci-discuss.net/os-2017-97/os-2017-97-AC1-supplement.pdf

---

## Author Comment (AC2) · 16 Apr 2018

The comments of the reviewer #2 are quoted below, the reply is given under each question.

Q1: The manuscript presents a very important topic in ocean color remote sensing and uses an extensive dataset combined with Monte carlo simulations. In my opinion the manuscript requires some revision to improve flow of text and aims of the study.

Reply: Thank you for appreciating my subject and for a number of detailed comments. I hope they will help me improve my paper.

Q2: Here are some aspects that could help improve the manuscript General questions after reading the manuscript i am wondering -would polarization be useful in winter

times assuming more days of overcast skies and diffuse light?

Reply: Remote sensing in the Baltic Sea area is very difficult in the winter season. This is due to very short days and a small number of sunny days. I hope that the polarization will be useful in the winter season. However, my article does not provide an analysis of the optical properties of the atmosphere in winter and therefore I am not able to give a judicious answer to this question.

Q3: -what are the uncertainties in your observed dataset?

Reply: The measurements were made using the ac-9 meter. Of course, these measurements are burdened with uncertainties that arise from two reasons. The first is multiple scattering inside the reflecting tube, and the second one is the imperfection of the mirrored walls of the tube.

Q4: -i am missing the statistical analysis of the data in Table 1 to show the 'seasonal vari- ability' of the dataset or at least the simulated dataset.

Reply: I agree, that such information is needed. Standard deviations are added to the table 1 after Sagan 2008.

Q5: In my opinion since the goal of remote sensing is to get Remote sensing reflectance an essential climate variable, i am wondering would it not be possible for your study to derive Rrs from the insitu (as- suming this will be your reference dataset) measurements assuming a Bb etc then you also do the same for simulated polarized light to estimate the Rrs and compare the two Rrs from simulation and insitu looking at differences over seasons

Reply: Thank you for that suggestion, it is interesting. But it goes beyond the subject of my work. Possibly I will do it in my next paper.

Q6: -the IOPs of the different water types would vary seasonally and i am wondering do you have any additional data to support some of the points you raise in the discussion that is measurements of the water constituents and backscattering maybe even from

satellite as a proxy. e.g Primary production enhanced or algal bloom could lead to higher scattering of the target region in the open water

Reply: Additional measurement results are included in Sagan's book (2008). This is a comprehensive analysis written as a habilitation dissertation (next academic degree after PhD in my country).

Q7: -what is the message from Table 2 how can one use such information? what are the implications of this study fidnings in advancing ocean color remote sensing

Reply: The data from table 2 allows one to recreate the function that describes the correlation between absorption to attenuation ratio and value of the DoP peak. Moreover there is answer to your question Q11.

Q8: additional comments Abstract line 1 -revise 'Polarization of light may be used to im- prove the remote colour sensing of sea water.', is it not something like 'Polarization of light leaving the ocean surface has the potential to improve quality of information retrieval from ocean color remote sensing (OCRS)'

Reply: Sounds better – done.

Q9: line 2 -'sun glints' do you mean 'surface reflected and sky glint'

Reply No, I do not. Sun glints are very bright flashing solar reflections, that can be seen above sea surface. They create silver area seen from high altitude and disturb the OCRS.

Q10: line 8 -'several years' can you put a number to this?

Reply: Yes measurements were performed for a total of four years. Done.

Q11: line 11 -'strong correlated' can you put some numbers to this as this is a qualitative statement

Reply: I added the sentence: The coefficient of determination R2 for different SZA and

wind speed varies between 0.906 and 0.996. These data are in Table 2.

Q12: Introduction line 16 -replace 'marine' with 'aquatic' to include all water types

Reply: Done.

Q13: paragraph 1 -there is an interesting paper by Brady might be of interest to you Brady et al 2015. Open ocean fish reveal an omnidirectional solution to camouflage in polarized environments. Science. 350(6263):965-969

Reply: I admit that this is a very interesting article. My congratulations to all authors. Added sentence: "Recent videopolarimetry measurements have shown that open-ocean fish species have higher camouflage abilities in polarized light than those that live nearshore, see Brady et al 2015."

Q14: paragraph 3 -replace 'colour remote sensing' with OCRS ->ocean color remote sensing

Reply: Done.

Q15: line 17 - what do you mean by this sentence 'Another factor that disturbs the colour remote sensing is surface reflected light that comes both from sky reflections and sun glints, that for some directions makes it difficult to gain a signal from water depth.' -is it not the case that surface reflected and sky glint affect OCRS –> because they are part of total water leaving radiance (glint + water constituents)?

Reply: Thats the clue. The water leaving radiance is the part of upwelling light that physically was in water (under the surface). One have to distinguish two components of upwelling radiance, namely "water leving" and surface reflected. The first one is scattered (single or multiple) by water constituents and hence it is affected by them, and the second is just reflected by the surface. See for example: Mobley 1999 Estimation of the remote-sensing reflectance from above-surface measurements or http://www.oceanopticsbook.info/view/overview_of_optical_oceanography/reflectances The water leaving radiance (its direction, spectrum and polarisation) is changed by

water constituents. That's why it contain useful information and that's why we want to measure its properties.

Q16: line 2 page 6 - 'after ?.' what does this mean?

Reply: Thank you. That's just a mistake – removed.

Please also note the supplement to this comment:
https://www.ocean-sci-discuss.net/os-2017-97/os-2017-97-AC2-supplement.pdf